# Caveolae-Associated Molecules, Tumor Stroma, and Cancer Drug Resistance: Current Findings and Future Perspectives

**DOI:** 10.3390/cancers14030589

**Published:** 2022-01-25

**Authors:** Jin-Yih Low, Marikki Laiho

**Affiliations:** 1Department of Radiation Oncology and Molecular Radiation Sciences, Johns Hopkins University School of Medicine, Baltimore, MD 21287, USA; mlaiho1@jhmi.edu; 2Sidney Kimmel Comprehensive Cancer Center, Johns Hopkins University School of Medicine, Baltimore, MD 21287, USA

**Keywords:** caveola, caveolin1, cavin1, stroma, cancer, drug resistance, p-glycoprotein

## Abstract

**Simple Summary:**

Cell membranes contain small invaginations called caveolae. They are a specialized lipid domain and orchestrate cellular signaling events, mechanoprotection, and lipid homeostasis. Formation of the caveolae depends on two classes of proteins, the caveolins and cavins, which form large complexes that allow their self-assembly into caveolae. Loss of either of these two proteins leads to distortion of the caveolae structure and disruption of many physiological processes that affect diseases of the muscle, metabolic states governing lipids, and the glucose balance as well as cancers. In cancers, the expression of caveolins and cavins is heterogenous, and they undergo alterations both in the tumors and the surrounding tumor microenvironment stromal cells. Remarkably, their expression and function has been associated with resistance to many cancer drugs. Here, we summarize the current knowledge of the resistance mechanisms and how this knowledge could be applied into the clinic in future.

**Abstract:**

The discovery of small, “cave-like” invaginations at the plasma membrane, called caveola, has opened up a new and exciting research area in health and diseases revolving around this cellular ultrastructure. Caveolae are rich in cholesterol and orchestrate cellular signaling events. Within caveola, the caveola-associated proteins, caveolins and cavins, are critical components for the formation of these lipid rafts, their dynamics, and cellular pathophysiology. Their alterations underlie human diseases such as lipodystrophy, muscular dystrophy, cardiovascular disease, and diabetes. The expression of caveolins and cavins is modulated in tumors and in tumor stroma, and their alterations are connected with cancer progression and treatment resistance. To date, although substantial breakthroughs in cancer drug development have been made, drug resistance remains a problem leading to treatment failures and challenging translation and bench-to-bedside research. Here, we summarize the current progress in understanding cancer drug resistance in the context of caveola-associated molecules and tumor stroma and discuss how we can potentially design therapeutic avenues to target these molecules in order to overcome treatment resistance.

## 1. Introduction

Small, vesicle, and “cave-liked” structures were discovered in the gallbladder epithelial cells in 1955 and were named “caveola intracellularis” or little caves [1]. These structures, which are 50–150 nm in size, were later named caveolae. They are predominantly expressed in cardiac muscle, certain continuous endothelial and epithelial cells, as well as adipocytes [2,3,4]. The caveolae were shown to play critical roles in cellular physiology and pathophysiology including lipid regulation, cellular signaling events, endocytosis, transcytosis of macromolecules, mechanosensing, cell adhesion, and migration [5].

Using electron microscopy, caveolae were described as smooth and to have a bi-polar orientation. Additionally, the presence of thin striations of structural caveola-associated coat proteins named as caveolin-1 (CAV1), caveolin-2 (CAV2), and caveolin-3 (CAV3) were described [6]. More recently, another family of caveolae-associated proteins, cavins, were discovered; today, the cavin family consists of four family members (cavin-1 to -4) [7].

The expression of CAV1 is critical for caveolae formation. Cells lacking CAV1 are devoid of caveolae, but these can be reconstituted by restoring the expression of CAV1 [8,9]. *Cav1* knockout mice demonstrate various physiological defects such as failure of renal calcium reabsorption leading to hypercalciuria and urolithiasis, vascular abnormalities, metabolic abnormalities, and the shortening of life span [10]. Apart from CAV1, CAVIN1 (also known as Polymerase-1 and Transcript Release Factor (PTRF) [11]) is another key molecule in caveolae formation [12]. Growing evidence suggests the importance of CAVIN1 in health and disease. *Cavin1* knockout mice have abnormal lung morphology and function and dysfunctions in glucose and lipid metabolism [13]. In humans, CAV1 and CAVIN1 mutations are associated with abnormalities in lipid levels and are present in congenital lipodystrophies [13,14].

CAV2, CAVIN2, and CAVIN3 were reported to play supporting roles in defining the morphology and regulation of caveolae. Although the expression of CAV2 is independent of caveolae formation, more abundant invaginations and more uniform caveolae formation were formed when CAV2 is co-expressed with CAV1 [15]. *Cav2* knockout mice have severe lung disorders, while no changes in the number of caveolae were observed [16]. Although expression of CAVIN2 alone does not alter the number of caveolae, its downregulation resulted in reduced expression of CAVIN1 and CAV1 and a consequent reduction in the caveolae number [17]. This suggests the interdependency between these three molecules and also tissue-specific loss of caveolae after CAVIN2 deletion [17]. The role of CAVIN3 is reported to be associated with CAV1 during the caveolae budding process along the microtubules and is greatly impaired in the absence of CAVIN3 [18]. Interestingly, in the *Cavin3* knockout mice there was no effect on caveolae numbers, body composition, or glucose tolerance, suggesting its function is redundant and the least critical of the cavins [19].

CAV3 and CAVIN4 are muscle-specific isoforms of the caveolin and cavin families [20,21]. Apart from its muscle-specific expression, CAV3 was shown to co-immunoprecipitate with dystrophin [22]. Loss of caveolae in sarcolemma was observed in *Cav3* knockout mice and together with insulin resistance, exclusion of dystrophin-glycoprotein complexes from the lipid rafts and abnormalities of the T-tubule system [23,24]. CAVIN4 was shown to be associated with cardiac dysfunction through mediating the Rho/ROCK signaling pathway [21]. Morphological and echocardiographic examination of *Cavin4* knockout mice showed attenuation of α1-adrenergic receptors induced by ERK1/2 activation and cardiac hypertrophy [25].

Apart from the roles of caveolins and cavins in cellular physiology, these molecules were also reported to be key players in driving an array of tumorigenesis, making caveolins and cavins viable targets for cancer treatment [6,13,26].

## 2. Cancer Drug Resistance Is Associated with Changes in the Caveolae

Resistance to cancer drugs remains a major challenge in the clinic. Drug resistance arises by multiple mechanisms including changes in drug transport and metabolism, mutation of the drug target, and activation of by-pass survival pathways often resulting from tumor heterogeneity allowing the escape and evolution of resistant cells [27,28,29]. These challenges are further complicated by multidrug resistance that results from increased expression of P-glycoprotein (P-gp), a transmembrane efflux pump, where cancer cells become simultaneously insensitive to an array of drugs [30]. Recent years of efforts in caveolae research have revealed the caveolae-associated molecules as regulators in cancer drug resistance. Here, we summarize the current achievements and breakthroughs in cancer drug resistance in the context of caveolae-associated molecules and how we can exploit these molecules to overcome drug resistance in order to deliver more promising drugs and treatments to the clinic.

Multidrug resistance is associated with marked changes in the plasma membrane composition such as an increase in cholesterol, other lipids, as well as CAV1 [31]. It was proposed that CAV1 may mediate the efflux of cholesterol, facilitating the delivery of drugs from intracellular compartments to the plasma membrane where the drug transporters reside [32]. On the other hand, overexpression of CAV1 may also reverse or attenuate the transformed phenotypes of drug-resistant cells [33], suggesting a complex and dual function of CAV1 in drug resistance. Recently, the role of CAV1 in cancer prognosis and therapy resistance, including chemo- and radiotherapy, has been reviewed [34,35]. Here, we discuss the caveolae, CAV1, and CAVIN1 and mechanisms of drug resistance both in the tumors and the surrounding stroma.

## 3. Mechanisms of Drug Resistance by CAV1 Associated with P-gp

As P-gp is a key protein in drug resistance, we first review known associations of CAV1 in drug resistance linked to P-gp (Figure 1). Caveolae were reported to be associated with P-gp, suggesting an important relationship between these two components in drug resistance [31,36]. P-gp, encoded by *ABCB1*, is a transmembrane ATP-dependent protein that transports xenobiotics unidirectionally out of the cell [37]. The expression of P-gp in tumor cells causes a reduction in intracellular drug concentration that affects the efficacy of a broad spectrum of cancer drugs [37]. In drug resistance, the expression of P-gp channels or other plasma-membrane ATPases tend to be upregulated to export cytotoxic drugs from the cancer cells [31].

Several studies have explored the expression and localization of CAV1 and P-gp. Characterization of CAV1 and P-gp in human bronchial epithelial cell layers using immunohistochemistry showed that P-gp was localized at the apical membrane and that these cells were also CAV1 positive [38]. Furthermore, caveolae were observed at the apical and basolateral membranes of the cells [38]. Additionally, the expression of CAV1 and P-gp were found to be co-localized in the luminal compartment of the endothelial cells in brain-tumor tissues [39]. In breast cancer, overexpression of *CAV1* in a doxorubicin-resistant breast cancer cell line (low CAV1 and high P-gp) resulted in downregulation of P-gp [40]. Subsequent studies in the same cell line showed that the overexpression of *CAV1* was associated with a reduced plasma-membrane cholesterol level abd increased plasma membrane fluidity, and it caused conversion of the cells from drug-resistant to drug-sensitive, possibly through the interaction between CAV1 and P-gp in the caveolae [41,42]. Another study showed that depletion of cholesterol using Methyl-β-Cyclodextrin (MBCD) removed P-gp from the caveolae and resulted in reduced P-gp function in vitro and intracellular accumulation of P-gp, which was reversed when cholesterol was replenished [43]. These early findings indicated a potential association between CAV1 and P-gp.

Several lines of evidence indicate that CAV1 is a negative regulator of P-gp and thus limits P-gp activity. The interaction between P-gp and CAV1 is essential for this activity, whereas a mutation in the P-gp CAV1-binding domain abolishes this association and increases P-gp transport activity [32]. Similarly, a mutation at lysine 176 to arginine (K176R) on CAV1 interferes with caveolae formation and breaks its interaction with P-gp [44]. Expression of K176R mutant CAV1 resulted in increased transport activity of P-gp and reduced the effectiveness of cytotoxic drugs in cancer cell lines [44]. In brain endothelial cells, CAV1 is also modified by phosphorylation on tyrosine 14 by Src tyrosine kinase, and this phosphorylation is necessary for its interaction and repression of P-gp [45] (Figure 1).

In contrast, in cancer cell lines, Src-mediated phosphorylation of CAV1 has been described to promote P-gp translocation to the caveolae and induce multidrug resistance [46]. Cbl-b, a member of the Cereblon family of E3 ubiquitin ligases, was found to reverse the resistance by causing degradation of Src, hence preventing the c-Src/CAV1-dependent translocation of P-gp into caveolae to overcome drug resistance [46] (Figure 1). This protective mechanism was supported by the finding that in patients receiving anthracycline-based chemotherapeutics Cbl-b predicted better disease-free survival in P-gp-positive patients [46].

In addition, in breast-cancer cells, Rack1 and Src tyrosine kinase regulate P-gp activity by modulating CAV1 phosphorylation. Rack1 acts as a signaling hub and mediates Src binding to P-gp, facilitating phosphorylation of CAV1 by Src and abolishing the inhibitory effect of CAV1 on P-gp [47]. These findings add to the emerging knowledge of the regulatory mechanisms upstream of CAV1 in promoting drug resistance and highlight some apparent differences in the models they have been tested so far. Notably, in normal cells such as the endothelia, P-gp activity supports expulsion of toxic compounds, whereas in cancer cells this propensity is further trimmed to enable escape from drug selection.

Recently, a study in cancer cell lines linked micro-RNAs (miR)-mediated regulation of CAV1 to drug resistance. MiR-103/107 was reported to be downregulated in a multidrug-resistant gastric carcinoma cell line, and its overexpression sensitized the cells to doxorubicin in vitro and in vivo [48]. Mechanistically, CAV1 and P-gp were verified as targets of miR-103/107, suggesting miR-103/107 target these two molecules independently to promote drug resistance [48]. An exosomal miRNA, miR-1246 was found to directly target and downregulate CAV1 in ovarian cancer cells, leading to an increase in the expression of PDGFRβ and P-gp (Figure 1). Overexpression of CAV1 and anti-miR-1246 treatment significantly sensitized ovarian cancer cells to paclitaxel and reduced tumor mutation burden in preclinical models [49]. This suggests that a drug-sensitive phenotype can be recovered by modulation of CAV1 expression.

## 4. CAV1-Linked Drug Resistance Mechanisms Independent of P-gp

Most normal epithelial cells are characterized by the presence of a low-level expression of CAV1. During cancer progression, the levels increase, although the mechanisms are not fully resolved. According to the Cancer Genome Atlas Program (TCGA) of human cancers, the amplification of the CAV1 locus is infrequent (<2%). The increase in CAV1 is, in most cancers studied so far, associated with phenotypes ascribed to advanced stages such as increased proliferation rates, invasion, metastasis and treatment resistance [34,50,51]. Hence, from a tumor perspective, the increased activity of CAV1 must be considered oncogenic. Yet, gain-of-function mutations of CAV1 are rare—in TCGA cancers there are infrequent (0.3%) CAV1 missense mutations scattered throughout the gene. In an experimental model of estrogen receptor-positive breast cancer, a proline 132 leucine mutation of CAV1 was associated with increased invasion, migration, and metastasis of cells in experimental models and was found in breast tumors [52,53]. However, this mutation has not been observed in further clinical material. Here, we address mechanisms associated with CAV1 in cancer drug resistance independent of P-gp (Figure 1). These have been widely studied across many cancer types, and for clarity we describe some of the key contributions separately for each cancer type.

### 4.1. Lung Cancers

Several studies report high expression of CAV1 in lung tumors and experimental models and association with drug resistance. Studies conducted in A549 human adenocarcinoma cells showed that CAV1 expression was upregulated after incubation with Paclitaxel [54], suggesting that continuous exposure to Paclitaxel can lead to a drug-resistant phenotype at least in part through upregulation of CAV1. A Paclitaxel-resistant derivative of A549 cells had undetectable expression of P-gp but had increased expression of CAV1 and an abundance of caveolae [54]. Knocking down *CAV1* in Taxol-resistant A549 cells induced G0/G1 cell-cycle arrest through a reduction in the level of cyclin D1 and activation of the Bcl-2/ Bax-mediated mitochondrial apoptosis in vitro and in vivo. Additionally, silencing *CAV1* reduced cell migration and invasion, attributed to AKT inhibition and downregulation of MMP2, MMP7, and MMP9 [55]. Similarly, A549 cells that were cultured at high concentrations of etoposide had increased expression of CAV1, although there was no correlation between CAV1 and P-gp distribution as they were localized in different membrane domains [56]. Another study in the A549 cells showed that bleomycin treatment resulted in cell-cycle arrest and cellular senescence accompanied by localization of multidrug resistance-associated protein (MGr1-Ag) in caveolae and formation of a complex between MGr1-Ag and CAV1 [57] (Figure 1). The implication of these findings is that MGr1-Ag can be inactivated by CAV1 to promote cellular senescence after bleomycin treatment. The activation of cellular senescence by bleomycin in lung cancer may provide a window of opportunity to resensitize lung-cancer cells to conventional chemotherapy.

Long-term nitric oxide exposure of lung-cancer cells has been shown to promote resistance to chemotherapeutic drugs. This observation was reported to be accompanied by upregulation of CAV1, followed by attenuation of cell death mediated by doxorubicin and etoposide. However, the observed drug resistance was reversible upon withdrawal of nitric oxide exposure [58]. These findings provide new insight where targeting CAV1 and nitric oxide in the tumor microenvironment could be beneficial in treating lung cancer.

In non-small-cell lung-cancer patients treated with gemcitabine chemotherapy, CAV1 immunopositivity was shown to be an independent prognostic indicator that predicts for poor disease-free survival and overall survival of patients [59]. A study in non-small-cell lung-cancer (NSCLC) cells reported that exposure to subtoxic cisplatin concentrations induces the production of hydrogen peroxide and reactive oxygen species that subsequently increased the level of CAV1 to promote an anoikis-resistant phenotype [60]. Furthermore, promoter methylation of CAV1 was associated with more favorable responses to taxane-platinum therapies in NSCLC [61].

### 4.2. Breast Cancer

Tamoxifen regulates CAV1 in MCF7 breast-cancer cells in a bimodal fashion. CAV1 increases early after the exposure followed by its downregulation after establishment of a stable tamoxifen-resistant phenotype [62]. Although the downregulation of CAV1 in MCF7 cells was not causal for the resistance, the finding indicates alternative regulation of CAV1 in the drug-resistant and sensitive cells [62]. Breast-cancer-resistance protein (BCRP) was reported to be downregulated following CAV1 knockdown, however the cellular localization of BCRP was not affected [63]. The knockdown led to reduced resistance to mitoxantrone (a BCRP substrate) but not to vincristine, which is a chemotherapeutic drug that is not extruded by BCRP [63]. Tyrosine 14 phosphorylation of CAV1 was reported to regulate paclitaxel-mediated apoptosis in MCF-7 breast cancer cells via downregulation of BCL2 and induction of p21 and p53 [64]. In a recent study in breast-cancer stem cells, it was reported that chemotherapy enhanced the expression of CAV1 expression in vitro and in vivo, and this was accompanied by co-overexpression of β-catenin and ATP-binding cassette subfamily G member 2 (ABCG2) [65]. Downregulation of CAV1 impaired the β-catenin/ABCG2 pathway through activation of glycogen synthase kinase 3 beta and inhibition of Akt, causing increased proteasomal degradation of β-catenin [65]. Additionally, using in vivo models, downregulation of CAV1 in breast-cancer stem cells also effectively impaired the tumorigenicity and chemoresistance [65], highlighting the importance of the CAV1-regulated β-catenin/ABCG2 signaling axis in chemoresistance of breast-cancer stem cells.

Trastuzumab emtansine (T-DM1) is a humanized monoclonal antibody-drug conjugate that has been approved for the treatment of metastatic breast cancers that are positive for human epidermal growth factor receptor 2 (HER2) [66]. However, resistance to this drug remains as a problem, and there are no biomarkers for T-DM1 drug resistance [67]. CAV1 was identified to be highly expressed in HER2-positive breast-cancer tissues. When CAV1 was overexpressed in HER2-positive breast-cancer cells, the cells were more sensitive to T-DM1 treatment. On the contrary, siRNA-mediated knockdown of CAV1 decreased the sensitivity of breast cancer cells to T-DM1 [68]. The expression of CAV1 was reported to mediate endocytosis and promote the internalization of T-DM1 into HER2-positive breast-cancer cells, thereby triggering apoptosis [68]. Using an alternative approach, metformin, a drug commonly used to treat diabetes, was found to upregulate CAV1 expression, enhance T-DM1 internalization, and promote the efficacy of T-DM1. When CAV1 expression was suppressed by shRNA, the effect of metformin-enhanced T-DM1 cytotoxicity was decreased [69]. This study suggested that metformin can be applied prior to T-DM1 treatment to improve the efficacy of T-DM1 by enhancing CAV1-mediated endocytosis [69]. Conversely, in a separate study, breast-cancer cells were found to internalize T-DM1 into intracellular CAV1-positive puncta and alter their trafficking to the lysosome. Instead of sensitization, this correlated with a reduced response to T-DM1 in a panel of HER2+ cell lines [70]. Further studies are required to reconcile these opposite findings to understand the role of CAV1-mediated endocytosis on T-DM1 and how this affects cellular sensitivity to the drug.

### 4.3. Renal Cancers

High expression of CAV1 confers resistance to doxorubicin-based chemotherapy in highly metastatic renal cancer cells. Knocking down CAV1 resulted in sensitization of the cells to doxorubicin and decreased the incidence of pulmonary metastasis in vivo [71]. In clear-cell renal-cell carcinoma (ccRCC), high CAV1 expression is associated with poor disease-free survival. Knocking down CAV1 inhibits cell migration and invasion, whereas overexpression of CAV1 promoted cell migration and invasion in ccRCC. Furthermore, the upregulation of CAV1 expression was shown to promote sunitinib resistance [72].

### 4.4. Sarcomas

The relevance of CAV1 tyrosine 14 phosphorylation has also been studied in rhabdomyosarcoma. Similarly to studies in other cancer cell lines, Src kinase was shown to phosphorylate CAV1 in human and mouse rhabdomyosarcoma cell lines. Enforced expression of CAV1 also resulted in increased phosphorylation of ERK and AKT [73]. This in turn contributed to enhanced cell proliferation, aggressive phenotypes, and resistance to cisplatin and doxorubicin, highlighting that inhibition of CAV1 phosphorylation may be a therapeutically valuable target to overcome drug resistance [73]. Similarly, high expression of CAV1 in Ewing’s sarcoma cells contributes to increased resistance to doxorubicin and cisplatin [74]. The resistance was reported to be mediated by PKC-α threonine 638 phosphorylation downstream of CAV1 that confers the sensitivity of the sarcoma cells to chemotherapeutic drugs [74].

### 4.5. Liver Cancer

A paclitaxel-resistant liver-cancer cell line was generated by exposing cells to increasing concentrations of paclitaxel and was found to associate with increased CAV1, fatty acid synthase (FAS), and P-gp expression. In addition, the cells were resistant to methotrexate, vinblastine, and doxorubicin but retained sensitivity to cisplatin. Downregulation of these three molecules resensitized the cells to paclitaxel [75]. Interestingly, CAV1 and FAS may interact, based on a report of their co-immunoprecipitation, suggesting a functional interaction between these molecules [75]. Their role in liver-cancer drug resistance compared to P-gp needs to be further defined.

### 4.6. Pancreas Cancer

CAV1 is overexpressed in pancreatic cancer, and high levels are associated with worse clinical outcomes, protumorigenic functions, and treatment resistance [76]. Further, CAV1 expression was found to directly correlate with nanoparticle albumin conjugate (nab) of paclitaxel sensitivity in pancreas-cancer and NSCLC cells and pancreas-cancer mouse xenograft models [77]. This study further suggested that CAV1 could be a predictive biomarker for albumin-conjugated therapies. This was further explored by testing the efficacy of altered therapeutic-administration schedules by first pretreating the cells with gemcitabine, which led to upregulation of CAV1 and increased uptake of nab-paclitaxel [78]. This strategy was found to increase the treatment efficacy and improve the survival benefit. Similarly, overexpression of CAV1 in a pancreatic-cancer cell line resulted in less-aggressive phenotypes and attenuated the chemoresistance to doxorubicin [79]. In contrast, knockdown of CAV1 sensitized pancreas-cancer cells to ionizing radiation concomitant with reduced expression of β1-integrin and Akt phosphorylation and activation of apoptosis by caspase-3 and caspase-8 [80]. The knockdown further enhanced the radiosensitivity of the pancreas cancer cells and increased numbers of residual DNA-double strand breaks [80]. This suggests that CAV1 expression may affect the sensitivity of cancer cells also towards radiation therapy.

### 4.7. Colorectal and Gastric Cancers

CAV1 expression is upregulated by rosiglitazone, a peroxisome proliferator-activated receptor-gamma (PPAR-γ) ligand in colon-cancer cells [81]. The activation is mediated by Src, EGFR, and the MEK1-ERK1/2 and p38 MAP kinase pathways, suggesting that complex regulatory pathways are involved [81]. CAV1 expression was also found to be high in methotrexate-resistant HT29 colon-cancer cells, and its downregulation by siRNA concomitant with overexpression of E-cadherin significantly reduced cell viability suggesting that these are potential targets for combination therapy in colon cancer [82].

Wingless-type MMTV integration-site family members (WNTs) are secreted glycoproteins involved in embryogenesis. Their dysregulation has been noted in cancers. In clinical specimens of gastric cancer, the expression of WNT6 positively correlated with the tumor stage and the nodal status and was negatively associated to the response to chemotherapeutic drugs including epirubicin, cisplatin, and 5-fluorouracil [83]. CAV1 enhanced the expression of WNT6, which increased the resistance to epirubicin and doxorubicin-mediated apoptosis. Furthermore, the activity of human WNT6 promoter was hyperactivated by epirubicin through CAV1-dependent binding of β-catenin to the proximal WNT6 promoter [83]. Collectively, these findings indicated that resistance to chemotherapeutic drugs in gastric cancer is mediated through upregulated expression of WNT6 and CAV1.

The expression of CAV1 is epigenetically silenced by methylation in colon- and breast-cancer cells [84]. Exposure to methotrexate or etoposide induces promoter demethylation, which increases transcription and expression of CAV1, mediated by ERK activation and reactive-oxygen-species production. Reactive oxygen species promote Src family kinase-dependent CAV1 phosphorylation that may lead to Rac1 and metalloproteinase activation leading to increased migration, invasion, and metastasis [84]. These findings suggest that anti-neoplastic drugs drive CAV1 expression, leading to aggressive and metastatic phenotypes, with the implication that non-lethal concentrations of cancer drugs may actually drive, rather than inhibit, tumor progression.

### 4.8. Glioblastoma (GBM)

In glioblastoma (GBM) patient sera, miR-1238 is exosomally secreted and higher than in healthy controls. Further, it confers temozolomide-resistance of GBM cells. CAV1 is a direct target of miR-1238 and is inversely correlated with miR-1238 expression [85]. The downregulation of CAV1 by miR-1238 led to activation of the PI3K-AKT-mTOR pathway in temozolomide-resistant GBM cells [85]. Furthermore, increased chemosensitivity induced by the downregulation of miR-1238 in the temozolomide-resistant GBM cells was reversed by CAV1 downregulation, suggesting miR-1238 controls CAV1 expression, leading to the drug resistance [85].

### 4.9. Prostate Cancer

Castration-resistant prostate cancers (CRPC) have high expression of CAV1 [86,87]. This was shown to be an independent risk factor for the occurrence and shorter recurrence-free survival time in patients with CRPC [88]. CAV1 induced the invasion and migration of CRPC cells by the activation of the H-Ras/phosphoinositide-specific phospholipase Cε signaling cascade in the caveolae [89]. Importantly, simvastatin, a 3-hydroxy-3-methylglutaryl coenzyme A reductase inhibitor, downregulates the expression of CAV1 in caveolae by blocking cholesterol biosynthesis, which in turn delays the progression of CRPC [89]. In vivo, a combination of anti-CAV1 antibody treatment along with dasatinib or sunitinib demonstrated significant tumor regression when compared to the drug treatment alone [90], suggesting the importance of drug combinations in overcoming treatment resistance.

Collectively, CAV1 affects mechanisms and pathways involved in drug resistance, some of which are related to P-gp, and many of which are linked to other intracellular signaling pathways. Overall, these findings affirm the pleiotropic effects of CAV1 on the cell physiology and pathophysiology (Table 1). There continues to be a high level of interest to define the molecular pathways of CAV1-mediated drug resistance in order to tailor suitable therapeutic strategies to overcome the treatment resistance.

## 5. Stromal CAV1 and Drug Resistance

Stromal cells abundantly express CAV1, and changes in CAV1 expression in the tumor stroma have been broadly reported. Loss of CAV1 in tumor stroma confers poor outcomes in prostate, breast, ovarian, esophageal, gastric, liver, and pancreatic cancers and melanoma [26,35,91,102]. Despite that, high expression of CAV1 in the stromal fibroblasts has also been described [103], and despite the fac that the expression appears to be context-dependent, the prevailing evidence points to an association between loss of stromal CAV1 and a worse disease outcome. At this time, it is not known whether the changes in stromal CAV1 are actively orchestrated by the tumor to reprogram the microenvironment or whether they represent adaptive changes of the stromal cells to the adjacent tumor. The extensive documentation of association of stromal loss of CAV1 indicates its value as a prognostic marker and highlights the severity of this alteration on the tumor phenotype.

Several experimental models detail the potentially associated perturbations that either support the tumor growth or resistance to therapeutic approaches (Table 1). Early studies, later confirmed using multiple oncogenes, showed that oncogene expression in fibroblasts was associated with a decrease in CAV1 [104]. In fact, *Cav1*-null mice are predisposed to multi-organ fibrosis, suggesting a major perturbation of the stromal component as a consequence [105]. The loss of Cav1 is sufficient to confer a cancer-associated fibroblast (CAF) phenotype with a change in gene-expression profile, altered deposition of extracellular-matrix components, thickening of the interstitial space, and fibrosis [106]. These stromal abnormalities also lead to epithelial hyperplasia and promote the growth of adjacent tumors [107,108]. Genetic ablation of *Cav1* increases the proliferation of primary and transformed fibroblasts via an increase in the MAPK signaling pathway [107]. Knocking down CAV1 increases the amount of cholesterol and testosterone in fibroblasts and promotes cancer-cell proliferation, primary tumor growth, and metastasis [100].

Loss of CAV1 in prostate-tumor stroma is a highly frequent and well-documented perturbation [52,100,101,109,110]. Further, the loss of CAV1 expression in the tumor stroma leads to the resistance of prostate epithelial cells to radiation [101]. As a mechanism, stromal fibroblasts were shown to respond to radiation by upregulating a p53 target gene and apoptosis inhibitor, TRIAP1, in a CAV1-dependent manner [110].

The loss of stromal CAV1 is a powerful predictor of poor clinical outcomes in breast cancer and was observed in independent studies [91]. It is associated with early recurrence, lymph-node metastasis, and resistance to tamoxifen, and it was observed across all types of breast cancers including the triple-negative breast cancers [91,92]. Furthermore, the expression of CAV1 is lower in lymph-node metastases than lymph nodes with reactive hyperplasia, indicating that a low expression of CAV1 in fibroblast-like stromal cells in lymph nodes correlates with high metastatic potential [111]. Low CAV1 expression in the stromal fibroblasts is mediated by oxidative stress, which increases the degradation of CAV1 by autophagy [26]. In breast-cancer tumor stroma, CAV1 expression is decreased, which was associated with increased expression of glycolytic enzymes, PKM2 and LDH [112]. This has been termed as the “reverse Warburg effect” indicating that loss of CAV1 drives a switch from oxidative phosphorylation to glycolysis, generating a microenvironment that drives the growth of the adjacent tumor.

The absence of CAV1 in the stromal cells conditions the environment also by paracrine signaling. The co-injection of Cav1-deficient dermal fibroblasts with melanoma cells is sufficient to recapitulate the tumor phenotype observed in *Cav1*-null mice as the *Cav1*-null fibroblasts promoted the growth of melanoma cells via enhanced paracrine cytokine signaling [113]. Interestingly, when breast-cancer cells that are sensitive to tamoxifen were co-cultured with tumor stromal cells, resistance to tamoxifen developed [99]. This suggests that tumor stroma may significantly influence drug resistance through reprogramming the metabolic-tumor microenvironment. A recent study by Kamposioras et al. showed that when genetically modified stromal cells were co-injected with tumor cells in a pancreatic cancer model [114], the tumor growth rate was accelerated, and resistance to gemcitabine developed when CAV1 was silenced [114].

In a phase-2 clinical trial of NSCLC, higher stromal CAV1 expression was associated with improved survival in patients who received nanoparticle albumin-bound (nab)-paclitaxel. This further indicates that CAV1 may be a relevant prognostic or predictive biomarker in NSCLC [96]. With the growing evidence that stromal CAV1 regulates treatment resistance, more research efforts are warranted to understand the role of stromal CAV1 contributing to treatment resistance and how to apply this knowledge during clinical decision making.

## 6. CAVIN1 and Drug Resistance

Similar to CAV1, genetic alterations of CAVIN1 in human cancers are rare, represented by less than 2% mutations and amplifications. The somatic mutation frequency is <1%, and most mutations are represented by missense alterations. These are often observed in cancer types with high mutation frequencies such as bladder and endometrial cancers and melanomas suggesting that they are likely passenger mutations. CAVIN1 expression is decreased during prostate-cancer progression [102,115,116] and is strongly correlated with that of CAV1 in prostate-cancer cells and tumors [117]. CAVIN1 levels are increased in GBM [118,119] and breast and pancreas cancers [93,120], while they are decreased in colorectal and lung tumors [121,122].

CAVIN1 expression has been linked with a response to chemotherapy (Table 1). The expression of CAVIN1 is upregulated in an adriamycin-resistant breast-cancer cell line along with the expression of P-gp, CAV1, and CAVIN2 [93]. The downregulation of CAVIN1 reduced the amount of lipid rafts on the surface of the adriamycin-resistant breast-cancer cells and reduced the drug-resistant phenotype, suggesting the importance of CAVIN1 in mediating drug resistance via the lipid rafts [93]. In GBM cells selected for resistance to imatinib and upregulation of P-gp, high CAVIN1 expression was linked with drug resistance. Furthermore, the expression of CAVIN1 and CAV1 were increased in relapsed GBM patients [94]. The silencing of CAVIN1 using siRNA increased the chemosensitivity of the resistant GBM cells to an array of drugs, including imatinib, etoposide, and temozolomide. Notably, the silencing of CAVIN1 also led to decreased expression of CAV1 highlighting the close connection between the two [94]. In silico analysis revealed upregulation of CAVIN1 in acquired lapatinib-resistant breast cancer [123]. Separately, CAVIN1 was identified as a resistance gene to phenylbutyrate, a histone deacetylase antagonist [124]. However, the biological significance of these observations remains to be further explored. CAVIN1 is upregulated in MAPK-inhibitor-resistant melanoma and is a significant biomarker for poor progression-free survival [97]. In experimental models, the overexpression of CAVIN1 led to MAPK inhibitor resistance, increased cell adhesion, and anchorage-independent growth [97].

## 7. Stromal CAVIN1 and Drug Resistance

CAVIN1 is highly expressed in normal prostate stroma but lost during cancer progression [102]. The decrease in stromal CAVIN1 and CAV1 correlated with reduced relapse-free survival, higher Gleason scores, and poor outcomes [100,102]. Recently, we investigated the role of stromal CAVIN1 in prostate cancer and showed using co-culture models of stromal cells and prostate-cancer cells that the depletion of CAVIN1 in the prostate stromal cells led to their inability to uptake lipids and hence caused lipid redistribution into the microenvironment [125]. The loss of stromal CAVIN1 activated inflammatory and CAF-like gene signatures and increased the secretion of cytokines and metalloproteinases such as MMP3, DKK1, and CSF1 [125]. Orthotopic implantation of the prostate-cancer cells with the stromal cells showed that CAVIN1 knockdown markedly increased the primary tumor lipid content and M2 macrophage infiltration and increased distant metastasis [125]. These findings show that the maintenance of the caveolae function, through CAVIN1, is essential for the lipid homeostasis and prevention of a shift towards an inflammatory microenvironment. It also suggests a close interaction between the cancer cells and stromal cells that converge through CAVIN1 by recruitment of inflammatory macrophages and promotion of a tumor-supportive microenvironment. Increased inflammatory signaling has also been observed in *Cavin1*-/-mice, and interestingly, these signatures are further augmented in mice receiving a high fat diet [126].

Based on our RNA-sequencing data, we observed a four-fold increase in P-gp expression after CAVIN1 knockdown in the stromal cells (ref. [125] and data not shown). This further supports the concept that loss of CAVIN1 not only impacts the lipid-regulatory inflammatory pathways but also P-gp expression. The aspect that stromal CAVIN1 could modulate drug resistance is intriguing. The increase in drug efflux into the microenvironment should translate to higher drug concentrations available to the tumor cells. Given the consequences of CAVIN1 on the formation of caveolae affecting lipid uptake, these findings raise the intriguing question whether there is a dietary component affecting drug resistance. These aspects are relevant to address in future studies as, to date, there are limited reports on the role of stromal CAVIN1 in drug resistance. Nevertheless, whether stromal CAVIN1 interacts with other lipid rafts or caveolae-associated molecules such as P-gp or other downstream signaling molecules to confer drug resistance remains to be elucidated.

## 8. Perspectives and Future Approaches

Efforts to understand caveolae-associated molecules in drug resistance have unearthed valuable descriptive and mechanistic knowledge on the roles of these molecules (especially CAV1) in promoting drug resistance (Figure 2) (Table 1). Therefore, it seems that targeting CAV1 and possibly CAVIN1 are relevant therapeutic goals to overcome drug resistance in cancer patients. Towards these goals, robust biomarkers are needed for stratification of patients into drug-responsive or non-responsive groups prior to treatments to enable selection of the most-effective therapeutic alternative for each patient. The implementation of CAV1 as such a marker has been explored. CAV1 has been proposed as a potential prognostic marker for radioresistance [51,127], and existing evidence points also to its prognostic role in cancer drug resistance. For example, the expression of CAV1 in oral cancer cells and tissues was reported to govern chemosensitivity to cisplatin and were suggested to have prognostic value [95]. Additionally, Chung et al. (2015) suggested that CAV1 may be a potential predictive marker to stratify HER2-positive breast-cancer patients to receive Trastuzumab emtansine [68]. Therefore, it is of importance to choose the most-appropriate treatments with knowledge of the resistance mechanisms present in both treatment-naïve tumors, and those arising during the course of the therapy, in order to provide the best, cost-effective regimens. Furthermore, the identification of individual patients who are responsive to specific treatments will reduce the treatment burden and the risk of using chemotherapeutic drugs that may promote cross-drug resistance.

It has been documented that CAV1 and CAVIN1 regulate the development and expression of the immune-regulatory system and molecules [115,128,129,130]. Moreover, the expression of immunomodulatory molecules, such as interleukin-6 and interleukin-8 in cancer cells, has been reported to be involved in drug resistance [131,132]. Although we are lacking a comprehensive outlook how knowledge on CAV1 and CAVIN1 could benefit cancer immunotherapies, it is reasonable to suggest that targeting these two molecules, especially in the tumor microenvironment, may change the immune landscape and aid in reprogramming the host immune system against drug-resistant cancer cells. This potential advantage may weaken the defense system of the drug-resistant cancer cells and improve the efficacy of chemotherapeutic drugs.

The tumor microenvironment is an exciting therapeutic target, and current progress continues to unravel how these minute spaces affect tumor growth and the therapy response. CAV1 and CAVIN1 are intimately connected to the regulation of the tumor microenvironment [91,103,133,134], and the tumor microenvironment is linked with regulation of cancer drug resistance [135,136,137]. This provides a new therapeutic opportunity whereby targeting CAV1 and CAVIN1 may modulate the tumor microenvironment enabling mitigation of drug resistance in cancer. This is supported by studies that have shown that CAV1 is a potential prognostic marker in the context of the tumor microenvironment [52,138,139].

Targeting two molecules involved in drug resistance at the same time using a combination therapy to overcome the resistance has been explored [140]. Since the expression of CAV1 is associated with P-gp, it is rational to target both molecules in parallel using two drugs or inhibitors. In this instance, the use of methyl-B-cyclodextrin (disrupting caveolae) or CAV1 inhibitory peptides and verapamil targeting P-gp may be an effective combination to overcome drug resistance. However, the delivery should be tumor specific in order to avoid systemic effects and may only work on solid tumors [141]. On the other hand, CAVIN1 may provide an entirely novel approach to mitigate drug resistance. This is based on the unique role of CAVIN1 affecting caveolae formation, cholesterol dynamics and, potentially, RNA synthesis [11,12]. Therefore, combining inhibitors that target CAVIN1 and cancer-drug metabolism may effectively shut down three routes at the same time (cholesterol dynamics at the plasma membrane, RNA synthesis, and the specific pathway that the other drug targets), and this could critically weaken the cancer cells and provide a viable path during treatment resistance.

Lastly, as conventional therapies such as radiation and chemotherapy remain as the choice of treatment for aggressive disease, it is important that cancer cells can be resensitized to these standard-of-care therapies. The emerging epigenetic therapies, through inhibition of epigenetic pathways may resensitize radioresistant prostate-cancer cells [142] and have applicability in reversing the stromal expression of the caveolae-associate proteins. Therefore, a combination of these strategies modulating the expression of CAV1 and CAVIN1 may provide a window opportunity to mitigate drug resistance in cancer cells.

## 9. Conclusions

Drug resistance remains a pervasive obstacle in cancer therapy. Extensive and intensive research is required to achieve new breakthroughs in understanding the underlying reasons in drug resistance and application of the new developments in the clinic. The role of CAV1 in cancer drug resistance was broadly explored, however more knowledge is required to exploit this concept in the clinical setting. In the future, targeting caveolae-associated proteins, especially CAV1 and CAVIN1, may unravel a new paradigm in the treatment of drug resistance in cancer.

## Figures and Tables

**Figure 1 cancers-14-00589-f001:**
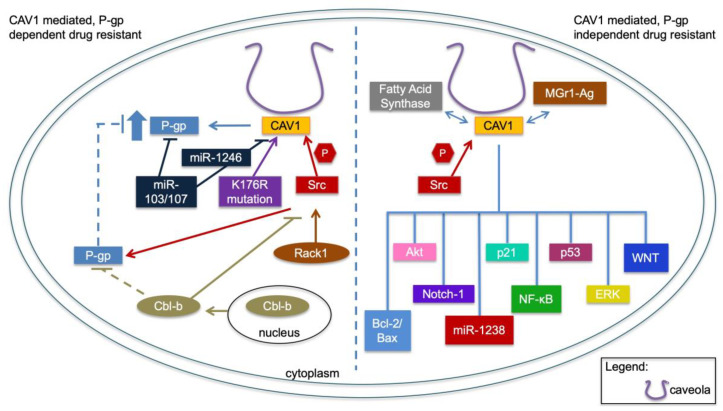
CAV1-mediated drug resistance. (**Left**) panel summarizes the mechanisms that are involved in drug resistance related to P-gp and CAV1. (**Right**) panel illustrates signaling pathways that are involved in CAV1-associated drug resistance independent of P-gp.

**Figure 2 cancers-14-00589-f002:**
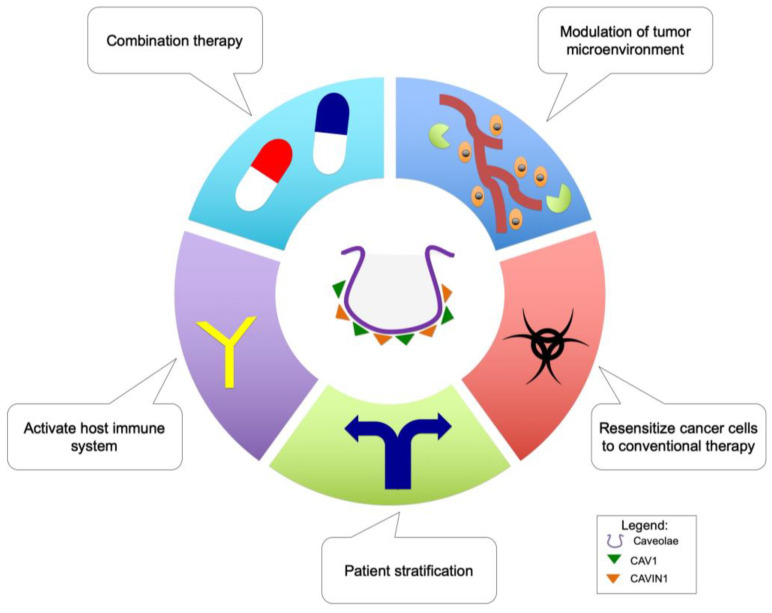
Therapeutic strategies to mitigate cancer drug resistance through targeting caveolae-associated molecules.

**Table 1 cancers-14-00589-t001:** Summary of CAV1 and CAV1N1 expression in tumor and stroma and their effect on drug resistance.

Cancer Type	Molecule	Tumor vs. Stroma	Expression Status	Effect on Drug Resistance
Breast	CAV1	Tumor	High	Knockdown of CAV1 resulted in downregulation of Breast cancer resistant protein (BCRP) leading to sensitivity to mitoxantrone [63] Downregulation of CAV1 in breast cancer stem cells promotes chemosensitivity [65] High CAV1 correlated with responsiveness to Trastuzumab emtansine in HER2 positive breast cancer cells [68] High CAV1 correlated with reduced response to Trastuzumab emtansine [70]
		Tumor	Low	Low CAV1 associated with doxorubicin resistance. Overexpression downregulated P-gp expression, resulting in a shift from drug resistance to drug-sensitivity [40,41,42,43]
		Stroma	Low	Association with tamoxifen resistance [91,92]
	CAVIN1	Tumor	High	Resistance to adriamycin [93]
Colorectal and gastric	CAV1	Tumor	High	Resistance to methotrexate [82] Resistance to epirubicin and doxorubicin [83]
Glioblastoma	CAV1	Tumor	Low	Resistance to temozolomide [85]
	CAVIN1	Tumor	High	Resistance to imatinib, downregulation of CAVIN1 sensitized cells to imatinib, etoposide and temozolomide [94]
Head and neck	CAV1	Tumor	High	Correlation with cisplatin resistance [95]
Liver	CAV1	Tumor	High	Resistance to methotrexate, vinblastine and doxorubicin [75]
Lung	CAV1	Tumor	High	Resistance to paclitaxel, etoposide, doxorubicin, bleomycin, gemcitabine and cisplatin [54,56,58,60]
			Low	Correlation to better outcomes with taxane-platinum therapies [61]
		Stroma	High	High stromal CAV1 expression associated with improved survival in patients who received nanoparticle albumin-bound (nab)-paclitaxel [96]
Melanoma	CAVIN1	Tumor	High	Resistance to MAPK inhibitor [97]
Ovary	CAV1	Tumor	High	Resistance to cisplatin [98]
Pancreas	CAV1	Tumor	High	Knockdown of CAV1 sensitizes to chemotherapies and ionizing radiation [76,79]
			Low	Uptake of nab-paclitaxel dependent on CAV1 [77,78]
		Stroma	High	Downregulation of stromal CAV1 in pancreatic cells promoted tumor resistance to gemcitabine [99]
Prostate	CAV1	Tumor	High	Decrease in CAV1 sensitizes to dasatinib and sunitinib [90]
				Resistance to antiandrogens [100]
		Stroma	Low	Resistance of prostate epithelial cells to radiation [101]
Renal	CAV1	Tumor	High	Resistance to doxorubicin and sunitinib [71,72]
Sarcomas	CAV1	Tumor	High	Resistance to cisplatin and doxorubicin [73,74]

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
