# Peer review of "Caveolae-Associated Molecules, Tumor Stroma, and Cancer Drug Resistance: Current Findings and Future Perspectives"

_cancers, 2022, doi:10.3390/cancers14030589_

Round 1

Reviewer 1 Report

 In the present review, the authors Here, Low & Laiho expose the current progress in understanding cancer drug resistance in the context of caveola-associated molecules and tumor stroma, and discuss how we can potentially design thera-peutic avenues to target these molecules in order to overcome treatment resistance. Caveola-associated molecules participates in the formation of caveolins and cavins. These are known to participate in numerous cellular events such as lipid regulation, cellular signaling events, endocytosis, transcytosis of macromolecules, mechanosensing, cell adhesion and migration among others. All these processes are modulated in tumors and in tumor stroma, and their alterations are connected with cancer progression and resistance to current treatments.

The manuscript is well written and clear and concise. However in order to strenght it, the authors could add a table summarizing the level of expression of CAV1 and CAVIN1 among the different type of tumors. Furthermore, it should be encouraged to add information about CAV-2, CAV-3 and CAV-4 in the different type of cancers exposed in the manuscript. All this suggestions will facilitate the comprehension to the readers. 

Author Response

We greatly appreciate the comments and encouragement by the Reviewer. In response, we proceeded to revise the manuscript. We are pleased that these revisions strengthened the manuscript and its main message.

  1. The reviewer recommended including a table summarizing the expression of CAV1 and CAVIN1 in tumors and the impact of this on drug resistance. We agree and are now providing a new Table 1 to address this. This is very supportive for the manuscript message and content.
  2. We have also revised some of the English language and verified proper formatting for genes and proteins.

Reviewer 2 Report

The authors provide a well written review on the expression and role of CAV1 and CAVIN 1 in cancer. Below I have noted a few recommendations which will benefit the  review.

In Fig 1: The up/down regulators are not clear, e.g. miR 1246 is known to down regulate CAV1 expression but not P-gp, however the image indicates miR 1246 also down regulates P-gp. Please can the authors carefully annotate the image.

Given the dual role of CAV 1 and CAVIN 1 in cancer and cancer therapy. It may be beneficial to provide the expression of these molecules in the different cancer cells/ stromal cells and provide if CAV 1 and CAVIN 1 is beneficial or detrimental in these cells/ cancer type. I believe it will provide a nice summary to the written text and may be a useful tool currently missing in other reviews on this subject.

Author Response

  1. We greatly appreciate the comments and encouragement by the Reviewer. In response, we proceeded to revise the manuscript. We are pleased that these revisions strengthened the manuscript and its main message.

    The reviewer recommended including a table summarizing the expression of CAV1 and CAVIN1 in tumors and the impact of this on drug resistance. We agree and are now providing a new Table 1 to address this. This is very supportive for the manuscript message and content.
  2. We have also corrected Fig. 1 according to the Reviewer 1 comments.
  3. We have also revised some of the English language and verified proper formatting for genes and proteins.